# Neonatal Calf Diarrhea and Gastrointestinal Microbiota: Etiologic Agents and Microbiota Manipulation for Treatment and Prevention of Diarrhea

**DOI:** 10.3390/vetsci11030108

**Published:** 2024-02-29

**Authors:** Emma Jessop, Lynna Li, David L. Renaud, Adronie Verbrugghe, Jennifer Macnicol, Lisa Gamsjäger, Diego E. Gomez

**Affiliations:** 1Department of Clinical Studies, Ontario Veterinary College, University of Guelph, 50 Stone Road East, Guelph, ON N1G 2W1, Canada; ejessop@uoguelph.ca (E.J.); ll937@cornell.edu (L.L.); averbrug@uoguelph.ca (A.V.); 2College of Veterinary Medicine, Cornell University, 602 Tower Road, Ithaca, NY 14853, USA; 3Department of Population Medicine, Ontario Veterinary College, University of Guelph, 50 Stone Road East, Guelph, ON N1G 2W1, Canada; renaudd@uoguelph.ca; 4Department of Pathobiology, Ontario Veterinary College, University of Guelph, 50 Stone Road East, Guelph, ON N1G 2W1, Canada; jmacnico@uoguelph.ca; 5Department of Population Health and Pathobiology, College of Veterinary Medicine, North Carolina State University, Raleigh, NC 27606, USA; lgamsjaeger@ncsu.edu

**Keywords:** dairy cattle, dysbiosis, enteropathogens, gastrointestinal microbiome, neonate

## Abstract

**Simple Summary:**

Diarrhea in newborn calves is a major global health concern, leading to significant illness and death. However, understanding the changes in the gut bacteria of calves with diarrhea and its causes still needs to be expanded. Various studies have shown that the gut bacteria in calves with diarrhea differs significantly from that of healthy calves of the same age. One crucial question is whether these bacterial changes contribute to, or result from, the gut inflammation caused by the pathogens associated with calf diarrhea. This review summarizes the current information on the alterations in the gut bacteria of calves with diarrhea and how the pathogens linked to diarrhea affect these bacterial communities. In human and dog medicine, strategies like pre-and probiotics, colostrum feeding, and fecal microbiota transplantation (FMT) have successfully treated and prevented gut diseases. While there is some information on using probiotics to prevent diarrhea in cattle, there is limited knowledge on employing these methods for treating calf diarrhea, including prebiotics or FMT. The second goal of this review is to provide an overview of the current knowledge about the use and efficacy of prebiotics, probiotics, synbiotics, colostrum feeding, and FMT can be used to treat and prevent diarrhea in calves.

**Abstract:**

Neonatal calf diarrhea is the leading cause of neonatal morbidity and mortality globally. The changes associated with the gastrointestinal microbiota in neonatal calves experiencing diarrhea and its etiology are not fully understood or completely defined in the literature. Several studies have demonstrated that the fecal microbiota of calves that experience diarrhea substantially deviates from that of healthy age-matched calves. However, one key question remains: whether the changes observed in the bacterial communities (also known as dysbiosis) are a predisposing factor for, or the consequence of, gastrointestinal inflammation caused by the pathogens associated with calf diarrhea. The first objective of this literature review is to present the current information regarding the changes in the fecal microbiota of diarrheic calves and the impact of the pathogens associated with diarrhea on fecal microbiota. Modulation of the gastrointestinal microbiota using pre- and probiotics, colostrum feeding, and fecal microbiota transplantation (FMT) has been used to treat and prevent gastrointestinal diseases in humans and dogs. Although information regarding the use of probiotics for the prevention of diarrhea is available in cattle, little information is available regarding the use of these strategies for treating calf diarrhea and the use of prebiotics or FMT to prevent diarrhea. The second objective of this literature review is to summarize the current knowledge regarding the impact of prebiotics, probiotics, synbiotics, colostrum feeding, and FMT for the treatment and prevention of calf diarrhea.

## 1. Introduction

Neonatal calf diarrhea is a major cause of mortality and morbidity, accounting for more than 50% of total deaths in calves [1]. This results in substantial economic and productivity losses in the cattle industry [2]. The estimated cost of enteric diseases at the national herd in Great Britain in 2003 was 11 million GBP per annum [3]. The expenses associated with cryptosporidium infection per calf is 60.62 EUR in Belgium, 43.83 EUR in France, and 58.24 EUR in the Netherlands [4]. Based on previous studies, the prevalence of diarrhea in dairy calves varies from 38.5% in the USA, 23% in Canada, 53% in Korea, and 58% in Iran [5,6,7]. Calf diarrhea can be caused by infectious agents, such as viruses, bacteria, and protozoa, with the main causative agents being bovine rotavirus group A (BRoV-A), bovine coronavirus (BCoV), *Salmonella* spp., *Escherichia coli (E. coli)*, *Clostridium perfringens* type C, and *Cryptosporidium parvum (C. parvum).* Enteropathogens that are primarily implicated in calf diarrhea can also be found in healthy calves; thus, their presence does not always indicate disease [7]. In most cases, diarrhea is multifactorial, and various enteropathogens can be involved simultaneously in the disease.

A key factor in the health and disease of calves is the gastrointestinal microbiota. The gastrointestinal microbiota is a diverse community of microorganisms, including bacteria, that act symbiotically to maintain gastrointestinal and host health. Specifically, the microbes in the gastrointestinal tract (GIT) supply nutrients to the host and enterocytes, regulate the local and systemic immune system, and assist in the morphological development of the intestines [2,8]. The key bacterial groups that compose the GIT of healthy calves have been well studied, but with inconsistent results. What is known is that the disruption, or dysbiosis, of the gut microbiota is associated with gastrointestinal disease, and that the re-establishment of healthy microbiota is essential for recovery [2,8]. However, it is still unclear whether dysbiosis occurs before or after pathogenic microorganisms invade the GIT. This narrative review aims to summarize the key bacterial populations of the “healthy” gastrointestinal microbiota in calves and microbial changes related to calf diarrhea and to describe novel treatment methods for calf diarrhea based on manipulating the gastrointestinal microbiota.

## 2. Methodology

The objective of this review was to collect literature on the current information regarding the changes in the fecal microbiota of diarrheic calves and the impact of the pathogens associated with diarrhea on fecal microbiota, as well as to summarize the current literature on the impact of prebiotics, probiotics, synbiotics, colostrum feeding, and FMT for the treatment and prevention of calf diarrhea. An electronic search was conducted using the following key words: dairy calf, dysbiosis, enteropathogens, and gastrointestinal microbiome. Articles published until the time of publication, 2023, were considered for use in this review. The articles in this review were collected from the following online databases: Agricola (https://www.nal.usda.gov/agricola, accessed on 15 October 2023), Google Scholar (https://scholar.google.ca/, accessed on 25 September 2023), PubMed (https://pubmed.ncbi.nlm.nih.gov/, accessed 1 October 2023), and Science Direct (https://www.sciencedirect.com/, accessed 12 January 2023). Information from each of the references was included in this review if they discussed etiology, pathophysiology, diarrhea treatment and preventative measures, and if the gastrointestinal microbiota was assessed in healthy and diarrheic individuals.

## 3. Etiologic Agents and Pathophysiology of Diarrhea

### 3.1. Bacteria

The bacterial pathogens associated with calf diarrhea include *E. coli*, *Salmonella* spp., *Clostridium perfringens (C. perfringens)*, and *Clostridiodes difficile*. Enteropathogenic strains of *E. coli* are the primary cause of diarrhea and mortality in newborn calves [9]. There are pathogroups that allow *E. coli* to be classified into enterotoxigenic *E. coli* (ETEC), Shiga toxin-producing *E. coli* (STEC), enteropathogenic *E. coli* (EPEC), attaching and effacing *E. coli* (AEEC), and enterohaemorrhagic *E. coli* (EHEC) [10]. Enterotoxigenic *E. coli* is the most prevalent cause of neonatal calf diarrhea, especially in calves younger than 4 days of age (Figure 1) [10]. The attachment of ETEC to epithelial cells occurs via the presence of fimbrial antigens, with K99 antigen and heat-stable toxin being predominant [11]. Heat-stable toxin-mediated ETEC attaches to the villi and crypts of the small intestine and ultimately leads to the upregulation of chloride secretion into the intestine, which promotes water secretion into the intestinal lumen and leads to the development of diarrhea [10,11]. Thus, ETEC does not produce gross or histopathological changes because the toxins do not damage the intestinal epithelium; therefore, the diarrhea is not hemorrhagic. In older calves, EPEC, STEC, AEEC, and EHEC can cause diarrhea. Unlike ETEC, these pathogroups cause damage to the intestinal epithelium, and diarrhea can be hemorrhagic. However, a direct link between these pathogroups and calf diarrhea is not well established, as laboratories rarely type the *E. coli* detected in the feces of diarrheic calves older than 5 days [12].

The predominant *Salmonella* spp. in calves is *S*. typhyimurium, which is associated with acute diarrheal disease [10] (Figure 1). The virulence of *Salmonella* spp. is attributed to its ability to invade the epithelial cells in the GIT and stimulate the release of inflammatory cytokines, subsequently inducing an inflammatory reaction [13]. Furthermore, *Salmonella* spp. multiply in the gastrointestinal lymphoid tissues and evade the host’s immune system [10]. These responses cause ulceration and destruction of the mucosa, resulting in the malabsorption of water and electrolytes and the loss of plasma proteins [13].

*Clostridium perfringens* is a bacterium associated with calf diarrhea (Figure 1), but its role as a main pathogen is still under debate [10]. *Clostridium perfringens* induces toxin-mediated cell necrosis, resulting in pores in the intestine, which causes an influx of solute and water, leading to malabsorptive and hypersecretory diarrhea [13]. First, *C. perfringens* is a commensal organism of the GIT of calves. All types found have the capability of producing toxins; however, Toxin A is most frequently detected in samples [12]. Therefore, detecting this bacteria in the fecal samples of diseased animals or detecting a toxin gene does not prove that the bacterium is causing the disease. Although some ELISA assays allow for the detection of Toxin A in intestinal contents, these assays cannot distinguish among the different types of *C. perfringens* [12].

*Clostridium difficile* produces several toxins, such as A, B, and binary toxin, which can result in diarrhea [14,15]. These toxins stimulate an increased secretion of electrolytes and water in the lumen of the intestine and severe ulceration of the intestinal mucosa [16]. However, it does not appear to play a pathogenic role in calves, as *C. difficile* has been detected in fecal samples from healthy calves, with a prevalence ranging from 2 to 60% depending on the source [10]. Furthermore, *C. difficile* has not been found to experimentally induce diarrhea in calves, and there are no reports linking this bacterium to specific diseases [17].

### 3.2. Viruses

The two main viruses implicated in calf diarrhea are bovine coronavirus (BCoV) and bovine rotavirus (BRoV) (Figure 1). Bovine coronavirus is an etiological agent involved in the pathogenesis of neonatal diarrhea, winter dysentery in adult cattle, and respiratory diseases in cows and calves [10]. It infects both the small and large intestines, leading to the destruction of villi and severe diarrhea [18]. Specifically, BCoV-infected intestinal epithelial cells die, slough off, and are replaced with immature cells, which causes the intestines to lose their capability of absorption and digestive enzyme secretion, causing hyperosmotic and malabsorptive diarrhea [18]. The spike proteins found on BCoV allow it to enter the host cells and are crucial to its pathogenesis [10].

Group A BRoV is a major cause of gastrointestinal infections in cattle [10]. Bovine rotavirus infects the mature, non-dividing enterocytes of the villi in the small intestine, causing damage and decreasing the small intestines’ absorptive capacity [11,19]. In addition, BRoV produces enterotoxin NSP4, which interferes with cellular homeostasis by elevating calcium ion influx from the movement of water in the intestine [11]. The release of NSP4 alters the movement of nutrients and water across the intestinal epithelium, resulting in hypersecretory diarrhea [10,11].

### 3.3. Parasites

*Cryptosporidium* spp. is endemic in cattle worldwide and is one of the most important causes of neonatal calf diarrhea [20] (Figure 1). In cattle, the four species of *Cryptosporidium* are *C. parvum*, *C. bovis*, *C. ryanae*, and *C. andersoni* [20]. There is a relationship between the age of the calves and the species of *Cryptosporidium* causing diseases [20], where *C. parvum* is associated with clinical disease in neonatal calves (<20 days of age), while *C. andersoni*, *C. bovis*, and *C. ryanae* infect the small intestine of weaned calves (Figure 1) [20]. *C. parvum* is a prevailing pathogen that infects neonatal calves worldwide, specifically in the second week of life [20].

Upon ingestion, *C. parvum* oocysts release sporozoites that invade the intestinal epithelium of neonatal calves [20]. A specialized feeder organelle allows *C. parvum* to obtain energy and nutrients from the calf without eliciting an immune response, which then allows for sexual and asexual reproduction, finally leading to infection and parasite shedding in the feces [20]. *C. parvum* often results in diarrhea because of damage to the intestinal epithelium, which decreases the absorptive capacity of the villi and increases intestinal permeability [21]. Specifically, *C. parvum* attaches to the epithelial cell villi, leading to villous atrophy and a decreased total absorptive surface area. Ultimately, this will cause malabsorption of fluid, resulting in diarrhea [11]. Prostaglandin-mediated anion secretion leading to hypersecretion of water into the lumen of the intestine is also proposed as a mechanism of diarrhea in calves with *C. parvum* infections [11]. Depending on the degree of epithelial damage and anion secretion, the severity of the clinical signs can vary from mild diarrhea to severe hemorrhagic diarrhea [11].

## 4. Development of the Gastrointestinal Microbiota in Healthy Calves

Before birth, the GIT of calves is generally considered to be a sterile environment, and microbial colonization begins immediately post-calving [22,23]. During birth, calves are exposed to the dam’s vaginal, fecal, skin, and mammary gland microbiotas, which begin colonizing the neonate GIT [22]. For instance, meconium samples obtained 6 h after calving contained bacteria normally present in the udder skin, such as *Leuconostoc* and Citrobacter [24]. Within the first hours of the calf’s life, colostrum is fed to provide nutrients and, more importantly, immunological factors that protect them from pathogenic microorganisms [25]. Beyond that, colostrum appears to modulate the colonization of the GIT with commensal bacteria during early life. Specifically, calves receiving pasteurized colostrum have an increased abundance of bacteria associated with GIT health, such as *Bifidobacterium*. However, colostrum-deprived calves have a high abundance of *Lactobacillus* and *E. coli* in the feces, which are related to GIT inflammation and disease [26,27].

During the first 3 days after calving, *E. coli*, *Clostridium*, and *Bifidobacterium* are the most abundant bacteria detected in calf feces (Figure 1) [28,29,30]. During the first 4 weeks of life, the gastrointestinal microbiota undergoes rapid changes, with increased abundances of *Bifidobacterium*, *Bacteroides*, *Faecalibacterium*, *Butyricimonas*, *Clostridium*, *Eubacterium*, and *Lactobacillus* [2,31,32,33]. These genera are necessary for calves to properly digest milk and will decrease in abundance as they transition to solid feed [31,33]. During the first 4 weeks post-calving, the prevalence of *Bacteroides* and *Lactobacillus*, two genera associated with the digestion of milk, increase [33]. However, as weaning begins, these genera gradually decline in abundance [33].

These findings indicate that the colonization of the GIT of healthy calves is associated with a shift from facultative (e.g., *E. coli*) to obligate anaerobic (e.g., *Bacteroides*, *Faecalibacterium*, *Clostridium*, *Bifidobacterium*) bacteria. This shift is also observed in human infants and foals [33,34,35,36,37,38,39]. This shift is associated with increased butyrate-producing bacteria, which aid in energy storage and gastrointestinal health [34,37]. In healthy calves, large populations of anaerobic bacteria and low oxygen tension are observed [30,31,32,33,34]. Short-chain fatty acids (SCFA), by-products of bacterial metabolism, aid in maintaining low oxygen tension as well as the concentration of nitrate within the intestinal lumen and mucosa [40,41]. The importance of this low oxygen tension within the intestinal lumen is to create a state of oxygen reduction, which results in an environment that is unfavorable to pathogens [42].

## 5. Changes in Gut Microbiota during Calf Diarrhea

Regardless of the causative agent responsible for the onset of calf diarrhea, various studies report significant changes in the bacterial communities of the gut microbiota during diarrhea compared to what is considered a “healthy” or “balanced” microbiota [8]. During diarrhea, there is a shift from obligate anaerobes to facultative anaerobes in the GIT, resulting in dysbiosis [43]. The abundance of *Faecalibacterium prausnitzii*, *Lachnospiraceae*, and *Ruminococcacea* bacteria associated with gastrointestinal health decreases significantly during calf diarrhea (Table 1) [33]. Concurrently, an increase in *Lactobacillus*, *Streptococcus*, and Enterobacteriaceae, especially *E. coli*, is observed (Table 1) [42,44]. The reason for this shift is still not completely understood, but several factors can contribute to these changes. Oxygen tension in the GIT is proposed to be a factor associated with the shift from anaerobic to facultative anaerobes [41,42,45]. In healthy calves, low oxygen tension creates a favorable environment for obligate anaerobes. However, during gastrointestinal inflammation, an increase in oxygen and reactive oxygen species in the GIT could offer an ecological advantage to facultative anaerobes [41]. In calf diarrhea, an increase in oxygen tension can be linked to the presence of blood and hemoglobin in the mucosa and lumen of the GIT due to inflammation [11,43]. Diarrhea caused by pathogens other than ETEC usually results in different degrees of watery intestinal contents, edema, hyperemia, ulceration, or hemorrhage of the intestine [11,43]. Thus, inflammation of the intestine could favor an increase in hemoglobin carrying oxygen to the mucosa and lumen of the GIT, leading to increased oxygen tension [42]. The increase in oxygen tension can also be explained by the production of oxygen species by neutrophils associated with respiratory burst (the release of oxygen species) during intestinal inflammation [11,46]. Therefore, additional studies are needed to investigate whether changes in oxygen tension within the intestinal mucosa and lumen are indeed associated with the microbial changes observed during calf diarrhea [43].

Alterations in the luminal pH of the intestines can also result in changes in the bacterial communities of the intestinal tracts of diarrheic calves, as pH affects bacterial growth and metabolism [43]. Specifically, during acute diarrhea, the calf’s fecal pH decreases as D- and L-lactate increase [43,47]. The increase in D- and L- lactate results from increased lactate-producing bacteria, such as *Lactobacillus* [43]. This is a cyclical process, and as luminal pH continues to drop, it generates favorable conditions for the continued growth of *Lactobacillus* species and other acid-stable bacteria. The continued growth of *Lactobacillus* results in further increased levels of lactate in the GIT that can potentially damage the intestine and reduce its ability to transport electrolytes in the intestines [47], contributing to hyperosmotic and malabsorptive diarrhea.

An increase in the abundance of bacteria from the Enterobacteriaceae family is consistently reported in diarrheic calves [48,49]. Dysbiosis associated with inflammation results in alterations in the metabolites available to and originating from bacteria in the GIT of calves, resulting in an environment that favors the growth of Enterobacteriaceae [2,8]. For instance, *Salmonella* spp. and *E. coli* benefit from the production of ethanolamine, lactate, glucarate/galactarate, 1,2, propanediol, succinate, and L-serine during dysbiosis [50,51,52,53]. Metabolomics studies have also noted that the amino acid composition in the lumen of the GIT is altered during inflammation. In calves, the concentration of fecal amino acids (e.g., serine, alanine, valine, isoleucine, glycine, and leucine) and the genes associated with the metabolism of various vitamins and carbohydrates decrease during diarrhea [8]. This change in amino acid availability in the lumen of the GIT could favor some facultative anaerobes to proliferate.

In summary, the main microbial alteration occurring in diarrheic calves is the shift from obligated anaerobes to facultative anaerobes. The exact cause of this shift is still to be determined. However, changes in oxygen tension in the lumen of the intestines and nutrient availability appear to play a key role in facilitating some taxa to proliferate more than others. Longitudinal studies investigating changes in the fecal microbiota before diarrhea are warranted to understand which microbial changes predispose calves to diarrhea. Similarly, studies investigating the fecal microbiota of calves during recovery from disease can contribute to identifying the microorganisms that play an important role in the resolution of gastrointestinal inflammation. Studies investigating microbial and metabolic changes in diarrheic calves could advance the current understanding of the mechanisms responsible for dysbiosis in calves.

**Table 1 vetsci-11-00108-t001:** Bacterial taxa enriched in the gastrointestinal tract of healthy and diarrheic calves during the first weeks of life.

Healthy calves	Taxa	Type	Reference
	*Faecalibacterium*	Anaerobe	[29,31,41]
	*Lachnospiraceae* spp.	Anaerobe	[31,41]
	*Ruminococcacea* spp.	Anaerobe	[29,31,41]
	*Prevotella*	Anaerobe	[31,54]
	*Butyricicoccus*	Anaerobe	[2,29,41]
	*Treponema*	Anaerobe	[2,54]
Diarrheic calves			
	*Escherichia coli*	Facultative anaerobe	[2,6,41,42,52]
	*Lactobacillus*	Facultative anaerobe	[6,41,54]
	*Streptococcus*	Facultative anaerobe	[29,41,52]
	*Fusobacterium*	Anaerobe	[29,55,56]

## 6. Microbiota Changes in Calves Infected with Specific Etiologic Agents

Few studies have addressed the question of microbial alterations in calves infected with specific pathogens, and most of the available information originates from outbreaks rather than well-controlled experimental studies.

### 6.1. Bovine Rotavirus

The effects of BRoV on the gastrointestinal microbiota were previously studied in a small number of diarrheic calves [54]. Infection with BRoV reduces the richness and evenness of the fecal microbiota and alters the bacterial membership (the taxa present in each community) and structure (the abundance of each taxon in a community) of the fecal microbiota [54]. Calves with rotaviral diarrhea had a lower relative abundance of Firmicutes and Bacteroidetes and a high abundance of Proteobacteria compared to their healthy counterparts. At the genus level, the genera *Escherichia*, *Clostridium*, and *Streptococcus* increased in BRoV-infected calves, while *Blautia*, *Bacteroides*, *Lactobacillus*, and *Coprococcus* decreased [54]. One limitation of this study was the small sample size of five calves in both the healthy and rotavirus groups [54]. More importantly, the study lacked the inclusion of a group of calves with naturally occurring diarrhea associated with other pathogens to determine whether the specific changes in the bacterial communities were caused by BRoV or another pathogen associated with the development of diarrhea [54]. Furthermore, it is still unknown whether the changes in the microbiota observed in calves with BRoV-induced diarrhea are the cause or consequence of the BRoV infection. Thus, longitudinal studies assessing the gastrointestinal microbiota before, during, and after recovery from diarrhea are warranted.

### 6.2. Enterotoxigenic Escherichia coli

Studies evaluating the effects of ETEC on the microbiota of calves are lacking. In pigs, ETEC-induced diarrhea is associated with a decrease in the Bacteroidetes/Firmicutes ratio, which aligns with the changes observed in other animal species such as dogs [57]. ETEC-induced diarrhea in piglets decreases the microbial diversity in the jejunum and feces and lowers the abundance of *Prevotella* compared to healthy counterparts [55,57]. ETEC in piglets is also associated with an increased abundance of *Lactococcus* in the jejunum and *Escherichia-Shigella* in the feces. Of interest, oral inoculation of piglets with ETEC failed to induce diarrhea in six piglets, suggesting that specific conditions including the pre-existing microbiota may aid in the facilitation or prevention ETEC infection in piglets [57]. Administration of vasoactive intestinal peptide (VIP) to piglets with ETEC-induced diarrhea reduced the length of the disease and the weight loss of the piglets [56]. The effects of VIP on pig diarrhea are associated with shifts in the GIT microbiota, suggesting that the modulation of the neuroendocrine–immune response can also modify the gastrointestinal microbiota, facilitating the resolution of intestinal inflammation [56].

### 6.3. Cryptosporidium Parvum

*C. parvum* is one of the main agents causing diarrhea in calves; however, little is known regarding the effects of this protozoal organism on the gastrointestinal microbiota [58]. Infection with *C. parvum* in calves results in a reduction in the microbial diversity, and this reduction is proportional to the amount of oocytes detected in the feces [59]. Furthermore, an increase in the fecal abundance of *Fusobacterium* is reported in diarrheic calves infected with *C. parvum* compared to uninfected calves or calves with BRoV diarrhea [58,59]. A high abundance of *C. parvum* and *Fusobacterium* is associated with the severity of diarrhea in calves with both microorganisms [59]. These results suggest that the proliferation of *Fusobacterium* could play an important role in the pathophysiology of diarrhea due to *C. parvum* in calves. However, experimental studies are required to determine the causes and effects of this association and its effects on calf health.

## 7. Alternative Approaches to Restore the Gastrointestinal Microbiota of Diarrheic Calves

Currently, there are many proposed methods to restore healthy microbiota in different species, including the use of prebiotics, probiotics, synbiotics, colostrum supplementation, and fecal microbiota transplantation (FMT). However, studies assessing the potential benefits of some of these approaches (i.e., colostrum supplementation, FMT) are limited in food-producing animals.

### 7.1. Prebiotics, Probiotics, and Synbiotics

The administration of prebiotics, probiotics, and synbiotics has been proposed as a method to treat dysbiosis by restoring microbial diversity and altering the disturbed intestinal microbiota in many digestive disorders, such as diarrhea, irritable bowel syndrome, inflammatory bowel disease, and ulcerative colitis in both calves and human neonates [9,59]. Prebiotics contain non-digestible nutrients that promote the growth of beneficial microbiota and aid in protecting the gut from potentially harmful pathogens [60]. Probiotics are groups of live, beneficial microorganisms that, when administered to patients sufficiently, will lead to overall health benefits for the host [60]. However, dead bacteria and their components can also exhibit probiotic properties [59]. Synbiotics contain a mixture of pre- and probiotics that are deemed advantageous to the host, promoting the growth and metabolism of beneficial bacteria [60].

Many prebiotics contain oligosaccharides, most commonly mannan and fructo-oligosaccharides, and they could deter harmful pathogens from colonizing as well as aid in reducing the severity of disease and diarrhea [61]. These non-digestible carbohydrates can be fermented by bacteria in the GIT and modify the bacterial communities and their function. The fermentation of prebiotics by bacterial communities results in the production of SCFAs [62]. The most-used prebiotics are oligosaccharides, which encompass various prebiotics, including β-glucans [62]. These oligosaccharides contain sugars that prevent Enterobacteriaceae, *E. coli*, and *Salmonella* from adhering to and colonizing in the intestinal epithelium [61]. However, the effects of prebiotics on the gastrointestinal microbiota and their associated mechanisms are poorly understood [60].

Probiotics are thought to competitively exclude pathogens, modulate enzymatic activities related to the metabolism of toxic substances, and aid in the production of fatty acids, which helps maintain energy homeostasis in peripheral tissues [63,64]. The mechanism through which probiotics improve gut health are still not exactly understood, but it is hypothesized that probiotics produce inhibitory substances, organic acids, hydrogen peroxides, and biofilms that reduce pathogen proliferation and promote healing of the intestinal lining [9,60,63]. Most importantly, in neonates, probiotics are associated with enhanced digestive and immune system development by increasing microbial diversity and species richness [9,61,65].

Studies have investigated the effects of probiotics as preventative measures for calf diarrhea. A meta-analysis assessing the effects of probiotics on the incidence of diarrhea and the intestinal microbial balance showed that the administration of probiotics containing lactic acid-producing bacteria reduced the incidence of diarrhea in calves fed whole milk [66]. However, this effect was only observed when a multistrain lactic acid bacteria probiotic was administered [66]. For diarrhea treatment, a single randomized clinical trial evaluated a multispecies probiotic containing *Pediococcus acidilactici*, *Enterococcus faecium*, *Lactobacillus acidophilus*, *Lactobacillus casei*, and *Bifidobacterium bifidum* used as a supportive treatment for calf diarrhea [67]. A total of 148 calves were enrolled in the trial, and all the calves received the probiotic at the onset of diarrhea and for three additional consecutive days. The mean time to diarrhea resolution was 5.1 and 5.9 days in the probiotic and control groups, respectively [67]. Although this study showed that a multispecies probiotic administered to the diarrheic calves reduced the duration of their diarrhea, the clinical relevance of this reduction is likely irrelevant, as no statistically significant differences were found [67,68].

One of the most frequently used probiotics that is administered to calves is live yeast, specifically *Sacchromyces cerevisiae.* In calves, the administration of probiotics containing yeast increases their grain consumption. This may be due to their ability to facilitate fiber digestion, which aids in gut development [69,70]. Yeast of *Saccharomyces cerevisiae* (SCY) origin is used in different food-producing animals to improve performance and health conditions [70]. In vitro studies have demonstrated the ability of yeast metabolites to inhibit pathogenic organisms and promote the growth of the commensal microbiota essential for producing volatile fatty acids [71]. In calves, SCY increases bacterial diversity and the abundance of bacteria from the family *Ruminococcaceae*, a butyrate-producing microorganism [72]. These findings highlight the potential benefit of SCY in modifying the gastrointestinal microbiota and the potential to accelerate recovery from diarrhea. However, the clinical and economic benefits of administering SCY to diarrheic calves are yet to be examined.

### 7.2. Colostrum Feeding

Appropriate administration of good-quality and a sufficient quantity of colostrum to neonatal calves decreases the incidence of diarrhea in the first weeks of the calf’s life [25]. Bovine colostrum, which is the milk produced by the dam immediately after parturition, has been found to contain many beneficial compounds such as nutrients, hormones, growth factors, and, most importantly, antibodies and immune factors essential in the first hours of life [73]. It has been proposed that antibodies, hormones, growth factors, oligosaccharides, and fatty acids contained in colostrum and spray-dried colostrum naturally optimize calf health and bolster immunity [73]. Administration of natural bovine colostrum after the first day of life has been shown to have beneficial effects on calves. For example, calves receiving natural colostrum for up to 14 days postpartum have less diarrhea and fewer antimicrobial treatments than control calves [74].

The benefits of spray-dried maternal-derived bovine colostrum replacer at the onset of diarrhea regarding calf growth and the duration and severity of disease in pre-weaned dairy calves have only been investigated in one study. This study showed that the administration of colostrum to diarrheic calves accelerated their recovery from diarrhea by 0.75 days and increased their average daily gain by 100 g/day compared to the control group [74]. Of interest, the administration of colostrum to diarrheic calves failed to alter the bacterial communities of the GIT [73]. However, in another study, daily supplementation with a high-quality bovine colostrum product ameliorated the clinical signs of calves experimentally infected with *C. parvum* and modulated the gastrointestinal immune response and microbiota to a pattern more similar to that of healthy, unchallenged calves [75]. Based on these results, supplementation of bovine colostrum appears to have beneficial effects both economically as well as physiologically for neonate calves. However, additional studies are required to further determine the effects of colostrum on the gastrointestinal immune system and gastrointestinal microbiota of calves.

### 7.3. Fecal Microbiota Transplantation

Fecal microbiota transplantation (FMT) is a procedure in which fecal matter from a healthy donor is administered into the intestinal tract of the recipient to directly change the recipient’s gastrointestinal microbiota [76]. The goal is to restore the “healthy” or “balanced” microbial communities within the recipient’s GIT to alleviate and facilitate recovery from disease [21]. The process of FMT was first described in human medicine in fourth-century China to treat diarrhea and gastrointestinal diseases [76]. Fecal transplantation is used to treat *C. difficile* infections in humans, with a success rate of 90% [77,78]. While the success of FMT in human medicine is mainly reported in cases of *C. difficile*, FMT has shown promising benefits for human patients suffering from irritable bowel disease, ulcerative colitis, chronic fatigue syndrome, and metabolic and cardiovascular disorders [78]. In dogs, FMT, in addition to standard treatment, accelerates the resolution of diarrhea in puppies suffering from parvoviral diarrhea compared with those receiving only standard treatment [21]. Additionally, FMT also offers protection against diarrhea in young pigs and prevents necrotizing enterocolitis in premature pigs [79,80].

In calves, little is known about the potential role of FMT for the prevention and treatment of neonatal diarrhea. Fecal transplantation from healthy donor calves to recipient diarrheic calves, regardless of the cause of diarrhea, decreases the water content in the feces [81]. Of interest, the fecal microbiota of calves receiving a FMT resembles that of the healthy donor following treatment, suggesting that transplanted bacterial communities are able to colonize the GIT and therefore facilitate recovery from diarrhea [81]. Recovery from diarrhea after a FMT is associated with an increased abundance of *Porphyromonadaceae* and *Prevotellaceae*, an increased concentration of SCFA, and a decreased fecal concentration of amino acids (alanine, leucine, valine, isoleucine, glycine, arginine, ornithine, and glutamic acid) [81,82]. These findings suggest that the bacterial metabolism of amino acids to synthesize proteins and other metabolites may be particularly important for recovery from diarrhea in calves. However, additional randomized clinical trials involving a larger number of calves from different farms are needed to establish the beneficial effects of FMTs on calf diarrhea. Similarly, there is still a need for further research to identify beneficial microorganisms in the feces that promote health, as well as the criteria to be considered a recipient or a donor calf [82].

## 8. Conclusions and Future Directions

Although many etiologic agents are involved and several pathophysiological mechanisms of diarrhea have been identified, the effects of the gastrointestinal microbiota in the health and disease of neonatal calves remain to be validated in multiple studies. While there are many etiological agents and causes of diarrhea, the overall impact on the microbiota and the role of the GI microbial community in disease is unclear. Several changes in bacterial composition occur during diarrhea, where there is a shift from strict anaerobes to facultative anaerobes and an increase in lactate-producing bacteria. An increase in lactic acid-producing bacteria appears to reduced pH and increase oxygen tension. Studies investigating these effects have been limited by the number of those enrolled in the study, single time point sampling, and lack of etiology as to the cause of the diarrhea.

Overall, determining whether changes identified in calves with diarrhea are the cause or the consequence of diarrhea, or whether the changes are pathogen-specific, is not possible without proper diagnostic testing. Furthermore, treatments for calf diarrhea that focus on modulating the microbiota hold many benefits and need to be further explored. Longitudinal studies evaluating the gastrointestinal microbiota before, during, and at the recovery from diarrhea are warranted. Similarly, studies characterizing the gastrointestinal microbiota in calves with diarrhea due to specific pathogens are needed.

## Figures and Tables

**Figure 1 vetsci-11-00108-f001:**
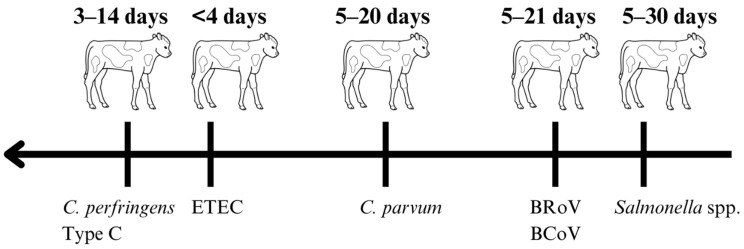
Age-associated pathogen distribution in calves with diarrhea. ETEC, enterotoxigenic *Escherichia coli*; *C. Parvum*, *Cryptosporidium parvum*; BRoV, bovine rotavirus; BCoV, bovine coronavirus.

## Data Availability

No new data were created or analyzed in this study. Data sharing is not applicable to this article.

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
