# Peer review of "Neonatal Calf Diarrhea and Gastrointestinal Microbiota: Etiologic Agents and Microbiota Manipulation for Treatment and Prevention of Diarrhea"

_vetsci, 2024, doi:10.3390/vetsci11030108_

Round 1

Reviewer 1 Report

Comments and Suggestions for Authors

The article presents the main agents causing neonatal diarrhea in calves and their effect on the microbial population in the gastrointestinal tract of these animals. The presentation of diarrhea-causing agents is adequate and reflects current knowledge on the topic. The article highlights the lack of knowledge about whether changes in microbial flora are predisposing factors or causes of neonatal diarrhea. Likewise, there are questions about whether factors that modulate intestinal microbial flora can act in the prevention or treatment of neonatal diarrhea in calves. The summary and introduction sections are complete, clearly show the objectives of the article and indicate what is intended to be addressed in the review. The microbial population of healthy animals is presented, as well as the main changes in the gastrointestinal microbiota and the main agents causing diarrhea. The use of intestinal microbiota modulators is presented, highlighting the lack of information on their the mode of action and their effects in the prevention and control of neonatal diarrhea. There are directions for future research, due to the knowledge gaps presented. There are no corrections in the text. Therefore, I consider that the review presented has scientific merit and can be accepted for publication in the form in which it is presented.

Author Response

Thank you very much for your comments. 

Reviewer 2 Report

Comments and Suggestions for Authors

Dear Authors,

Please see the attached file for the comments of this review.

Comments on the Quality of English Language

Author Response

Please see the attached word document, thank you.

Reviewer 3 Report

Comments and Suggestions for Authors

Neonatal calf diarrhea is a major global problem. The impact of gastrointestinal microbiota changes during diarrhea and their relationship with causative pathogens are not fully understood. This literature review addresses this gap, focusing on microbial shifts in diarrheic calves and the role of pathogens. Additionally, it explores the use of pre- and probiotics, colostrum feeding, and fecal microbiota transplantation for treating and preventing calf diarrhea. Despite their success in humans and dogs, limited information exists on their efficacy in calves. The review aims to summarize current knowledge on the use of these interventions in managing neonatal calf diarrhea.

This is a well-organized review of the literature. The following are minor concerns.

1. There are no figures or tables. Organizing the various studies with effects and references would be beneficial. Also, differences in bacteria taxa due to oxygen tension might be worthy of a diagram to increase clarity. 

2. On Page 5, second full paragraph, the list of amino acids should have only a single "and", and valine is listed twice and one should be removed.

3. On page 6, when discussing ETEC-induced diarrhea in piglets, I suggest the addition of a study showing how a gut neuropeptide, vasoactive intestinal peptide (VIP), when administered with ETEC (88) ameliorated not only diarrhea and weight loss but also the shift in the gut microbiota (Modulatory Effects of Vasoactive Intestinal Peptide on Intestinal Mucosal Immunity and Microbial Community of Weaned Piglets Challenged by an Enterotoxigenic Escherichia coli (K88), Xu C. et al. Plos One 2014 Aug. 7;9(8)). 

Author Response

(The authors gave the same response as above.)

Round 2

Reviewer 2 Report

Comments and Suggestions for Authors

Dear Author,

Please see the attached file for review report of your manuscript.

Author Response

Please see that attached word document. Thank you. 
